# Changes in Cell Morphology and Actin Organization in Embryonic Stem Cells Cultured under Different Conditions

**DOI:** 10.3390/cells10112859

**Published:** 2021-10-23

**Authors:** Younes F. Barooji, Kasper G. Hvid, Irene Istúriz Petitjean, Joshua M. Brickman, Lene B. Oddershede, Poul M. Bendix

**Affiliations:** 1Niels Bohr Institute, University of Copenhagen, 2100 Copenhagen, Denmark; farhangi@nbi.ku.dk (Y.F.B.); kasper.gr.hvid@gmail.com (K.G.H.); I.IsturizPetitjean@tudelft.nl (I.I.P.); 2The Novo Nordisk Foundation Center for Stem Cell Biology, 2200 Copenhagen, Denmark; joshua.brickman@sund.ku.dk

**Keywords:** actin cytoskeleton, super-resolution microscopy (STORM), embryonic stem cells, primed embryonic stem cells, micro-rheology, cell culturing, optical tweezers

## Abstract

The cellular cytoskeleton provides the cell with a mechanical rigidity that allows mechanical interaction between cells and the extracellular environment. The actin structure plays a key role in mechanical events such as motility or the establishment of cell polarity. From the earliest stages of development, as represented by the ex vivo expansion of naïve embryonic stem cells (ESCs), the critical mechanical role of the actin structure is becoming recognized as a vital cue for correct segregation and lineage control of cells and as a regulatory structure that controls several transcription factors. Naïve ESCs have a characteristic morphology, and the ultrastructure that underlies this condition remains to be further investigated. Here, we investigate the 3D actin cytoskeleton of naïve mouse ESCs using super-resolution optical reconstruction microscopy (STORM). We investigate the morphological, cytoskeletal, and mechanical changes in cells cultured in 2i or Serum/LIF media reflecting, respectively, a homogeneous preimplantation cell state and a state that is closer to embarking on differentiation. STORM imaging showed that the peripheral actin structure undergoes a dramatic change between the two culturing conditions. We also detected micro-rheological differences in the cell periphery between the cells cultured in these two media correlating well with the observed nano-architecture of the ESCs in the two different culture conditions. These results pave the way for linking physical properties and cytoskeletal architecture to cell morphology during early development.

## 1. Introduction

The biomechanical properties of cells allow them to sense and react in response to their physical environment, for example, by sensing the rigidity of the extracellular matrix. The mechanical properties, both of the cell and its environment, play an important role in cell organization, migration, and differentiation. During development, the physical properties of cells change drastically in terms of viscoelasticity and morphology, with the cell generally changing stiffness, as it progresses through development [1,2]. Furthermore, the rigidity of the environment has been shown to be crucial in differentiation, for regulation of the stem-ness of cells, and for the development of organoids [3]. The viscoelasticity of cells is also important in disease, for example, in cancer, where invasiveness has been linked to the cells’ stiffness and adaptability to extracellular matrix stiffness [4,5]. As the exact mechanisms and effects of both mechanical properties and mechano-sensing of cells during differentiation and development are not yet well-understood, it is important to map and understand the changes in physical properties of cells during differentiation and to correlate this with mapping of the differential expression of transcription factors. This would provide a deeper understanding of the interplay between mechanical and biochemical cues during differentiation.

The overall viscoelasticity and cytoskeletal organization of early embryonic stem cells (ESCs) and differentiated cells has been investigated, and cells have been found to undergo dramatic changes in their cytoskeletal organization and physical stiffness as they exit from pluripotency [1,2,6,7,8]. However, little information exists about the local sub-cellular viscoelastic properties of ESCs and the associated changes in the cytoskeletal structure as they differentiate. The actin cortex in ESCs has been imaged and shown to be regulated via a complex interplay between Arp2/3, formin, and capping proteins, while myosin II was proven to be only minimally involved [8,9].

Embryonic cells are known to exhibit dramatic changes in morphology during the first few days after fertilization. Early embryonic development can be modeled in vitro using ESCs, pluripotent cell lines derived from the mammalian embryo. Naïve ESCs are derived by the ex vivo expansion of the inner cell mass (ICM) of the blastocyst and can be cultured in a variety of conditions. As the ICM is in the process of segregating the embryonic epiblast and extra-embryonic endoderm lineages, ESC culture conditions can be varied to trap states resembling these lineages [10,11,12,13]. ESCs have traditionally been cultured in the presence of the cytokine LIF (Leukemia Inhibitory Factor) alongside serum, here termed Serum/LIF media. Under these relatively undefined conditions, ESCs contain populations of cells that are dynamically primed for both epiblast and endoderm differentiation [13]. Relatively homogeneous cultures can be obtained in these defined conditions that include small molecule inhibitors of GSK3 and MEK (2i) along with LIF, here termed 2i/LIF media or just 2i. Cells cultured under these conditions homogeneously express transcription factors associated with the epiblast lineage and also contain a sub-population that co-expresses these factors alongside determinants of extra-embryonic development. As a result, these cells approximate development identities resembling the early epiblast and inner cell mass of the peri-implantation blastocyst [11,14,15]. These cells exhibit more homogeneity than cells grown in serum media supplemented with LIF [11,13,16,17] and exhibit a remarkably different morphology when grown on a substrate [18]. To our knowledge, no study has quantitatively investigated the structural and physical properties of cells cultured under these two widely used culture conditions.

Here, we investigate mouse embryonic stem cells (ESC), which recapitulate cells from the developing embryo; however, they can also be cultured and expanded in vitro and studied in detail at the sub-cellular level. To address cell states that recapitulate aspects of early development, we culture ESCs in two well-described stem cell media conditions: (i) a complete media based on LIF and fetal bovine serum (FBS) and (ii) a media known as 2i/LIF. Cells grown under both conditions mimic cells from the mouse blastocyst, with the cells cultured in 2i better approximating early stages of development (Figure 1A,B). As these in vitro cultures reflect changes that occur in vivo as a result of morphogenetic changes and the physical segregation of cell types, the biomechanical properties of these cells are clearly of interest. In this study, we aimed to describe whether, on the single cell level, there would be differences in either the actin cytoskeletal structure or viscoelastic properties of cells grown in either 2i or Serum/LIF, respectively, representing cells in two cognate developmental states.

To address these questions, we used both confocal imaging and super-resolution fluorescence imaging of the actin structure in cells grown in 3D in small colonies. Confocal imaging was used to characterize the 3D shape of the cytoskeleton in ESC colonies grown under different culture conditions. Confocal imaging gives information on the morphology of individual cells in the 3D cluster as well as the overall distribution of actin in the cell, while it has a relatively good penetration depth in the axial direction; however, the optical resolution restricts it from being able to resolve the detailed structure of the dense network of filaments. To overcome this limitation and to examine the nanoscale structure of subcellular actin filaments, we implemented stochastic optical reconstruction microscopy (STORM) (Figure 1D and Appendix A), using the stochastic blinking of fluorophores linked to actin and individually detecting and localizing these molecules with nanoscale precision, thus mapping out the structure of the actin network [20,21,22,23].

To correlate the observed changes in cytoskeletal architecture with the micro-rheological properties of the individual cells, we probed the viscoelastic properties of live cells with sub-cellular resolution by optical trapping endogenously occurring lipid granules inside the cells. This method allows for precise quantification of the viscoelastic properties of the cytoplasm at the sub-cellular level [5].

## 2. Materials and Methods

### 2.1. Cell Culture

E14Ju mouse embryonic stem cells (ESCs) [12] were maintained in either full stem cell media with serum and LIF (Serum/LIF) or serum-free 2i/LIF media (2i), for no more than 25 passages, and were regularly tested for mycoplasma.

Serum/LIF medium consisted of GMEM (Sigma-Aldrich, 2860 Søborg, Denmark) supplemented with 10% Fetal Bovine Serum (FBS), 2 mM L-Glutamine, 1 mM Sodium Pyruvate, 0.1 mM 2-mercaptoethanol, 0.1 mM Non-Essential Amino Acids, and 1000 units/mL LIF (DanStem, Copenhagen, Denmark).

2i medium consisted of a 1:1 mix of DMEM F/12 (Gibco, Thermo Fisher Scientific, 4000 Roskilde) and Neurobasal Medium (Gibco, Thermo Fisher Scientific, 4000 Roskilde), supplemented with N2, B27, 100 μM 2-mercaptoethanol (Sigma-Aldrich, 2860 Søborg, Denmark), 1000 units/mL LIF (prepared in house bythe group of JMB, DanStem), 3 μM Chir99021 (Sigma-Aldrich, 2860 Søborg, Denmark), and 1 μM PD0325901 (Sigma-Aldrich, 2860 Søborg, Denmark).

Cells were grown on flasks (Corning) coated with 0.1% gelatin (Sigma-Aldrich, 2860 Søborg, Denmark) in 37 °C incubators containing 5% CO_2_ and were passaged with DPBS (Sigma-Aldrich, 2860 Søborg, Denmark) and Accutase (Sigma-Aldrich, 2860 Søborg, Denmark).

### 2.2. Cell Fixation and Staining

For confocal and STORM imaging, cells were plated on fibronectin-coated #1.5H thickness 8-well glass bottom slides (Ibidi, Gräfelfing, Germany). Fibronectin coating was performed by covering the glass slides in a solution of DPBS with 10 μg/mL human fibronectin (EMD Millipore, Burlington, MA, USA) for at least 2 h. Cells were then plated at a density of 20,000 cells/cm^2^ and left growing in specific culturing media (as described above) overnight.

For fixation, before performing confocal and STORM microscopy, each well was washed once with DPBS before adding 4% paraformaldehyde in PBS for 10 min. After fixation, cells were again washed and stored in DPBS at 5 °C until staining.

Before staining, cells were permeabilized using 0.1% Triton X-100 (Sigma-Aldrich, 2860 Søborg, Denmark) in PBS for 15–30 min. After washing in PBS, blocking was performed by incubating the sample with 1% BSA in PBS for 30 min at room temperature. Cells were then stained with Alexa Fluor 647 Phalloidin (Thermo Fisher Scientific, 4000 Roskilde at a 40× dilution of the stock solution for 20 min at room temperature. The cells were again washed in PBS before finally staining chromatin with the DAPI stain NucBlue Fixed Cell ReadyProbes Reagent (Thermo Fisher Scientific, 4000 Roskilde.

### 2.3. Confocal Microscopy 

Fixed and stained cells were imaged on a Leica SP5 confocal microscope (Leica Microsystems, Wetzlar, Germany) using a 63× water-immersion objective (NA = 1.20, Leica).

### 2.4. Micro-Rheology

Micro-rheological measurements were carried out using a tightly focused laser beam, an optical trap, implemented in the confocal microscope. The trapping laser had a wavelength of 1064 nm and was coupled in through the side port; the setup is described in Andersen et al. [24]. For these experiments, the cells were kept alive in a perfusion chamber with the lower coverslip coated by fibronectin. As a tracer particle, we used an endogenously occurring lipid granule; such particles are abundantly present throughout many cell types [5,25,26]. The laser was focused on a single granule located at the position of interest as detected by bright field imaging. The forward-scattered laser light was collected by a high numerical aperture oil immersion condenser and imaged onto a quadrant photodiode (QPD) located close to the back focal plane of the condenser. Data from the QPD were analyzed by a custom-made Matlab code based on the software published in Hansen et al. [27] and the methods described in Selhuber-Unkel et al. [26].

Briefly, the time series were Fourier transformed, and for each time series, the exponent, *α*, describing the scaling properties of the power spectrum in the frequency interval 300–3000 Hz, was extracted. In our experiments, the laser was operated at low laser powers, and this frequency interval was well below the corner frequency characterizing the stiffness of the trap; also, in this frequency interval, thermal fluctuations dominate over non-equilibrium activity within living cells [28]. For anomalous diffusion, such as the thermal fluctuations of a tracer particle within a crowded cytoplasm, the mean squared displacement (MSD) of the tracer particle scales with time-lag (*τ*): $MSD\left(\tau \right)\;\sim \;\tau^{\alpha }$, where *α* is the scaling exponent [25]. For 0 < *α* < 1, the movement is sub-diffusive [29], and within this regime, the lower the value of *α*, the more elastic the environment and the closer to 1, the more viscous the environment. For pure Brownian motion, *α* = 1, and 1 < *α* is a sign of super-diffusive motion. For tracer movement in a viscoelastic medium, the power spectrum scales with frequency, *f*, as:(1)$$P_{x}\left(f\right)\equiv \;\langle {\left|\tilde{x}\left(f\right)\right|}^{2}\rangle \propto \;f^{-\left(1+\alpha \right)}.$$
Hence, the scaling exponent, *α*, characterizing the viscoelastic properties of the medium, can be found by fitting Equation (1) to the power spectrum [5,26] in an appropriate frequency window.

We used the Stoke–Einstein equation, $G\left(f\right)=\frac{1}{6\pi rɣ\left(f\right)}$, to extract the storage,$\;G^{\prime }\left(f\right)$, and loss moduli, $G^{\prime \prime }\left(f\right),$ from the time series based on the framework presented in [30,31,32]:(2)$$G^{\prime }\left(f\right)=\frac{1}{6\pi r}\frac{{ɣ}^{\prime }\left(f\right)}{{ɣ}^{\prime }{\left(f\right)}^{2}+{ɣ}^{\prime \prime }{\left(f\right)}^{2}}$$
(3)$$G^{\prime \prime }\left(f\right)=\frac{-1}{6\pi r}\frac{{ɣ}^{\prime \prime }\left(f\right)}{{ɣ}^{\prime }{\left(f\right)}^{2}+{ɣ}^{\prime \prime }{\left(f\right)}^{2}}$$
Here, r is the radius of the particle inside the medium. ${ɣ}^{\prime }\left(f\right)$ and ${ɣ}^{\prime \prime }\left(f\right)$ are the imaginary and real parts of the response function of medium due to the particle’s fluctuations. Based on the fluctuation–dissipation theorem, the imaginary part of the response function is linearly related to the frequency and power spectral density, P(f):(4)$${ɣ}^{\prime \prime }\left(f\right)=\frac{\pi f}{K_{B}T}P\left(f\right)$$
where $K_{B}T$ is the thermal energy.

The real part of the response function can be calculated numerically using a Kramers–Kronig relation:(5)$${ɣ}^{\prime }\left(f\right)=\frac{2}{\pi }P\int_{f^{\prime \prime }=0}^{f^{\prime \prime }=\infty }df^{\prime \prime }\frac{f^{\prime \prime }{ɣ}^{\prime \prime }\left(f^{\prime \prime }\right)}{{f^{\prime \prime }}^{2}-f^{2}}$$

A couple of issues should be taken into account while applying this method. For a particle trapped inside the medium, the storage modulus contribution from the optical trap, $G_{trap}^{\prime }$, should be subtracted from calculated results [33]. In our experiment, $G_{trap}^{\prime }$ was extracted for a 0.5 μm trapped bead in water, which is a purely viscus medium ($G_{trap}^{\prime }=10.6\pm 0.5\;\mathrm{Pa}$), using the same laser power and equipment settings as in the cell experiments. In the experimental measurements, a finite maximum measurement frequency of 22 kHz was employed. Hence, care needs to be taken while interpreting at higher frequencies, as the finite sampling frequency will cause errors in the high-frequency region in the calculation of ${ɣ}^{\prime }\left(f\right)\;$using Equation (5). However, the error is negligible for frequencies $f<f_{Nyquist}/10$ [34], and data are therefore reliable in this frequency region. In our experiments, as the QPD was operated at 22 kHz, the shear moduli were extracted for f<1100 Hz (see Figure 6E,F).

This analysis provides quantitative values for both the loss and storage moduli, thus supporting the conclusion based on analyzing the power spectra

### 2.5. STORM Imaging and Analysis

STORM imaging was based on a custom-made total internal reflection (TIRF) Olympus IX83 inverted microscope (Olympus, Tokyo, Japan), equipped with a 150 × 1.45 NA oil-immersion objective (Olympus, Tokyo 163-0914, Japan), an EMCCD camera (Hamamatsu, Hamamatsu City 430-8587, Japan), a Z drift compensation system (IX3-ZDC2, Olympus, Tokyo 163-0914, Japan), and a motorized TIRF module (Olympus, Tokyo 163-0914, Japan). To enable STORM imaging, we individually fiber-coupled a 640 nm laser (Toptica Photonics, Gräfelfing, Germany) and a 405 nm laser (IX3-ZDC2, Olympus, Tokyo 163-0914, Japan) to the TIRF module and independently set both illumination angles to a TIRF penetration depth of 100–200 nm for imaging. More details of the setup can be found in the Appendix A.

To induce stochastic blinking of the Alexa Fluor™ 647 fluorophore, imaging was performed in a PBS-based imaging buffer based on the dSTORM buffer described by van de Linde and others [23] containing 100 mM MEA (Sigma-Aldrich, 2860 Søborg, Denmark), 0.6 mg/mL Glucose Oxidase (Sigma-Aldrich, 2860 Søborg, Denmark), 10% (*w/v*) Glucose (Sigma-Aldrich, 2860 Søborg, Denmark), and 60 μg/mL Catalase (Sigma-Aldrich, 2860 Søborg, Denmark) in PBS. The pH of the buffer was adjusted to between 7.5 and 8.5 by the addition of KOH.

Time-lapse imaging of blinking events was performed at 20 ms exposure time for 10,000–30,000 images. The exposure time is chosen such that a sufficient signal can be collected, but without obtaining multiple blinking events from the same location. Individual single molecule blinking events were then localized in the time-lapse images using the ThunderSTORM [35] plugin in Fiji [36], and drift was corrected using the cross-correlation function in ThunderSTORM.

### 2.6. 3D STORM Imaging

3D STORM imaging was performed as described above, with the important difference that a cylindrical lens was inserted in the optical pathway before the camera. This lens produced astigmatism in the detected emission from single-molecule blinking events, elongating them in either the vertical or horizontal direction depending on their z-distance from the focus plane [37]. The elongation was compared to the elongation from a control experiment where surface-fixed beads were scanned while gradually moving them through the focus in the z-direction. This allowed for 3D localization of blinking events using ThunderSTORM and the elongation–correlation dataset.

### 2.7. Image Analysis 

All quantifications were performed using custom written Matlab programs Matlab (Mathworks, Natick, MA, version 2018a).

The area and the aspect ratio (the ratio between major and minor axis) of adherent cells were calculated using Matlab (v. 2018a) and imageJ (v. 1.51h) software.

The contact angles shown in Figure 2C were calculated as the angle between the substrate (horizontal) and the edge of the cell colonies. For each colony, the angle was measured along the entire circumference of the colony and averaged. The contact area (as depicted in Figure 3E) was measured as the total area covered by actin for individual cells detected within the TIRF volume.

For quantifying actin cable orientation (as shown in Figure 5), STORM images were filtered using an implementation of a Gabor filter to enhance the spatial resolution of linear structures in images [38,39] (see Appendix A). After image enhancement, the local orientation angles of actin filaments in every pixel were extracted based on the structure tensor method using the freely available ImageJ plugin OrientationJ (v. 2.0) [40,41]. Figure 5A,B shows the color-coded angle distribution for two cells cultured in Serum/LIF and 2i media, respectively. The polar histograms of actin orientation, from Figure 5A,D, are shown in Figure 5C,D. The distribution of actin orientations indicates that actin filaments are predominantly orientated along the cell’s major axis for a cultured cell in Serum/LIF and distributed randomly for a cell in 2i. This observation can be quantified by an orientation order parameter, S=⟨cos(2θ)⟩, where *θ* is the local angle between the filaments in a given pixel and the direction of the cell’s major axis [42,43]. Order parameters, *S*, for the cells as depicted in Figure 5A,B were extracted for 16 cells in each medium (Figure 5E).

### 2.8. Real-Time Deformability Cytometry

To assess the overall stiffness of cells, we use real-time deformability cytometry, where cells are pumped through a channel constriction in a microfluidic device and experience deformation by pressure gradients and shear stresses, see Appendix A [44,45]. Cells were passing through a 20 µm channel cross-section. To optimize the success of the experiments, the diameter of the studied cells should be between 20–95% of the channel size.

When pumped into the chip, cells initially passed two rows of pillars to prevent cell debris from blocking the flow through the channel. There are two inlets entering the chip containing buffer (sheath) and the sample inlet containing the cells, see Appendix A. The buffer is named Cell Carrier and is a phosphate-buffered saline solution with <1% methyl cellulose.

Inside the narrow channel, the deformed cell is illuminated with a pulsed and high-power light-emitting diode, and a complementary metal oxide semiconductor (CMOS) camera is used for detection. The flow cytometry arrangement combined with real-time analysis of the cell shape allows for measurement at rates of up to 1000 cells per second.

## 3. Results

### 3.1. Geometry of Stem Cell Colonies Depend on Culture Condition

To perform confocal imaging of colonies of cells in three dimensions, we labeled fixed cells for F-actin and with a nuclear stain (Figure 2A,B). Actin was observed to mainly localize in the cell cortex both in cells cultured in Serum/LIF and in 2i, however, with important differences: (i) cells cultured in Serum/LIF have a higher number of filopodia and actin protrusions along the surface than those cultured in 2i, (ii) colonies of ESCs cultured in 2i grow in rounded relatively high dome-like colonies, as has previously been observed [46], whereas cells cultured in Serum/LIF media grow in flat and more spreading colonies of cells, as is apparent from the x-z projection (Figure 2A,B).

At the colony edges, the contact angles of cells adhering to the substrate under the two culturing conditions were significantly different. The contact angles, defined as the angle between the substrate and colony edge and averaged over the circumference of the colony (Figure 2C), were found to be *θ* = (115 ± 3)° for cells grown in 2i, and *θ* = (69 ± 9)° for cells grown in Serum/LIF. Hence, cells cultured in Serum/LIF media seem to maximize their contact with the surface by spreading out toward the substrate, whereas cells in 2i tend to minimize their contact to the substrate interface while the colony forms a dome-like structure.

### 3.2. Nanoscale Architecture of Actin Depends on Culture Condition

To investigate the nanoscale organization of F-actin, we performed STORM imaging at depths of up to a few hundred nanometers using TIRF illumination (Figure 3). With its higher resolution than confocal imaging, STORM imaging revealed that cells cultured in Serum/LIF have concentrated actin filaments in dense stress fibers and bundles that are mainly located on and along the circumference of the cell. Actin is also clearly visible in the abundantly present filopodia. Within the central region of Serum/LIF cells, we did not observe connected filaments within the TIRF volume, indicating a lack of connected filaments close to the surface. In comparison, actin in clustered 2i cells (as shown in Figure 3B) is more evenly distributed throughout the cell cortex at the substrate interface (within the TIRF volume).

To compare the spreading of cells grown under the two different culturing conditions, we quantified the contact area of each cell with the substrate. This was defined as the total area of cells visible within the TIRF volume (Figure 3E). We found that ESCs grown in Serum/LIF media exhibit significantly larger spreading on the surface than cells grown in 2i, thus corresponding well with the more spread-out and flatter geometry of colonies grown in Serum/LIF than in 2i (as visible in Figur 2). Additionally, as quantified in Figure 3F, cells grown in Serum/LIF have a significantly higher aspect ratio and hence exhibit a more elongated shape than cells grown in 2i.

STORM imaging of colonies of cells grown in 2i revealed clear gaps of a few microns at the surface plane between cells (Figure 3A). More irregular and fewer clear gaps are also visible between cells in a colony grown in Serum/LIF (Figure 3B). However, colonies grown in 2i have wider gaps, which appear to result from the more spherical shape of single cells and a larger contact angle formed with the surface by each of the cells cultured in 2i. These gaps could not be resolved by confocal imaging under the same conditions, indicating that actin polymers in individual cells grown in 2i are curving inward toward the center of the cell at the substrate (as indicated in the schematic drawing in Figure 6A). To obtain higher 3D resolution than is achievable by confocal microscopy, we investigated the structures by 3D STORM (Figure 4). For cells grown in 2i media, we did indeed see the actin curving away from the surface at the edges of the cell colony. This is in contrast to the situation for cells grown in Serum/LIF, where actin bundles can clearly be seen close to the substrate and pointing toward the edges or in the protrusions of the cell (Figure 4A). This is consistent with the presence of dorsal stress fibers in primed cells, attaching to the substrate at the leading edges of the cell [47,48]. For cells grown in 2i media, on the other hand, we do not see evidence of dorsal stress fibers; instead, an interconnected mesh of actin without any clear directions or bundles is observed (Figure 4B).

### 3.3. Nanoscale Organization of Actin at the Surface Depends on Media Conditions

In order to quantify the seemingly disorder in the orientations of filaments in 2i, we used 2D STORM data, and implemented a Gabor filter to emphasize and connect the STORM point detections into filaments. The filaments were then color-coded based on their orientation respective to the major axis of the cells (Figure 5). For each culturing condition, we quantified the orientation order parameter S=⟨cos(2θ)⟩, where *θ* is the angle between local fiber orientation and the overall cell orientation (Figure 5C). Actin filaments in stem cells cultured in 2i were found to be significantly more disordered than filaments in Serum/LIF cells (*p* < 0.01). The order parameters, *S*, for the cells, as depicted in Figure 5A,B, were found to be 0.23 and 0.05, respectively. The same analysis was performed for 16 cells in each medium, and the average result shows a significant difference in the order parameter: for cells cultured in Serum/LIF, the average was found to be <S> = 0.19, whereas for cells cultured in 2i, <S> = 0.04 (Figure 5D).

### 3.4. Viscoelastic Properties of Cells under Different Culturing Conditions

A significant difference in the actin structure of ESCs was observed under the two different culturing conditions, in particular, in areas close to the cell periphery; we next set out to probe the viscoelastic properties inside the cells, but close to the periphery. More precisely, using optical tweezers, we quantified the viscoelastic properties in a frequency interval of 300–3000 Hz where non-equilibrium processes have been shown to be negligible [28] and material properties are probed at timescales relevant for the dynamics of cytoskeletal elements. In this frequency interval, the positional power spectrum, *P_x_(f)*, scales with the scaling exponent, *α*, which carries information on the viscoelastic properties as outlined in Materials and Methods.

Figure 6C shows an example of a positional power spectrum from an endogenous lipid granule located at the cellular periphery of an ESC cultured in Serum/LIF, hence, a situation resembling the one shown in Figure 6B ii. For each culturing condition, 26 individual granules were trapped and their scaling exponent fitted and plotted in the boxplot shown in Figure 6D. For cells cultured in 2i, we found an average scaling exponent of *α* = 0.48 ± 0.11, while for cells cultured in Serum/LIF, we found an average value of *α* = 0.40 ± 0.15. In both cases, only granules close to the cell periphery were investigated. First, a D’Agostino and Pearson normality test was performed on the two distributions to validate normality; second, a Student’s *t*-test was performed on the two distributions, which returned a value of *p* = 0.03. Hence, on a 0.05 percent significance level, the two distributions would likely be independent. In conclusion, we find that close to the cellular periphery cells cultured in Serum/LIF are more elastic than cells cultured in 2i, which are more viscous. We additionally performed an analysis to extract the storage and loss modulus from the data as shown in Figure 6E,F, see Materials and Methods. This analysis showed that the storage modulus was higher for Serum/LIF cultivated cells for all investigated frequencies (Figure 6F), in accordance with results from the power spectral analyses. These data agree well with the observations revealed by STORM imaging that significantly denser and straighter actin filaments are present along the periphery of cells cultured in Serum/LIF than in cells cultured in 2i. We note that other structures such as intermediate filaments (e.g., nestin) could be partially responsible for the observed changes, but this remains to be studied in future work. Averaging over the entire cell, we observed no difference in viscoelastic properties between the cells cultured under the two cognate conditions; only at the periphery was a difference observed.

## 4. Discussion

Confocal imaging of ESCs cultured in Serum/LIF or 2i revealed clear differences in the morphology both of single cells and of the colonies they form. Cells grown in 2i had a more spherical shape and formed dome-like cell clusters. Cells grown in Serum/LIF, corresponding to a slightly later pluripotent state, showed pronounced spreading on the surface and formed monolayers when cultured on a surface.

Closer inspection of the actin structure by STORM imaging showed characteristic differences in the nanoscale structure of the actin cytoskeleton between cells cultured in 2i and cells cultured in Serum/LIF. Cells cultured in Serum/LIF were observed to spread along the surface by flattening and extending out actin protrusions with dorsal stress fibers firmly connecting to the substrate. ESCs cultured in 2i, on the other hand, mainly organize their actin in a cortical mesh with very few radial actin structures protruding from the cell and with less reinforcement along the cellular periphery than observed for cells cultured in Serum/LIF. The rounded shape of the individual cells cultured in 2i, as well as for the colonies they form, could be a result of low affinity for substrate attachment.

We probed the viscoelastic properties in the area close to the cells’ periphery, where we observed the significant geometrical differences in actin nano-architecture between the two culturing conditions. These results demonstrated a small difference at the cell periphery between the two culturing conditions, with the cells grown in Serum/LIF, which also displayed the most abundant actin structures along the cell periphery, being more elastic than cells cultured in 2i. It is reasonable that areas with more abundant actin polymers are more elastic, and this corresponds well with similar types of viscoelastic measurements performed on *S. pombe* yeast cells, which also demonstrated a decrease in elasticity upon actin disruption [25].

The mechanism behind the morphologies found for the two culture conditions remains to be investigated in detail. However, both mechanical and biological mechanisms must be at play. Recently, similar shapes, as found here, were explained through an active wetting mechanism in which cellular assemblies formed 3D organoids or 2D monolayers in response to the relative magnitude of intracellular forces and traction forces [49]. The analogy of passive wetting of droplets is, however, not justified since the mechanisms at play are fundamentally different in the two cases, but the concept of active wetting provides a useful framework for tissue spreading.

Biologically, the thick cortical actin in cells grown in 2i, lying adjacent to the plasma membrane, is likely to form tight cell junctions in 3D cell clusters by cell–cell adhesion through e-cadherin, thereby reinforcing the observed 3D dome-like structure. The question remains whether the actin structure could control the cell’s ability to differentiate by, e.g., regulation of the plasma membrane tension. The tension in membranes was recently found to regulate the fate of stem cells through tension regulation of endocytosis [50,51]. Our results show that stem cells already in the early stages of differentiation begin to show distinct viscoelastic properties at the cell periphery, although these differences are still much smaller than at later stages in development [2]. ESCs grown in Serum/LIF media have abundant and dense actin fibers along the cellular periphery and more elastic material properties close to the cell periphery than cells grown in 2i. An interesting finding related to the mechanics of the cell surface was reported in Bergert et al. [51]. Inhibitor withdrawal from 2i media caused cells to decrease their membrane tension drastically as they transition toward differentiation, and overexpression of the actin–membrane linking protein ezrin forced cells to remain in a naïve pluripotent state [50,51]. Here, we compare ESC cultures that resemble early epiblast and ICM-like identity to cells in serum/LIF, where cells actively progress in and out of the earliest stages in differentiation [12]. These are both defined and stable stages in our media conditions, whereas the studies reported in [50,51] report on the changes that occur as cells transition from the naïve stage. Hence, the cells in our study do not exit pluripotency, but we compare cells that are trapped deep in pluripotency to cells that can reversibly enter the earliest stages of differentiation, and as a result, membrane cortex adhesion should be preserved in all cases. Moreover, we focus on the actin structures located inside the membrane (not on the membrane itself); the presence of a denser cortical actin network does not necessarily exclude lower membrane tension. As described in Bergert et al. [51], during the transition, the membrane tension decreases from a reduction in membrane–cytoskeleton linkage, and the transition could be inhibited by artificially linking the membrane to the cytoskeleton.

We measured significant differences between the scaling exponent, α, characterizing the viscoelastic properties of the cytoplasm of cells cultured in 2i/LIF or Serum/LIF culture conditions, see Figure 6. Moreover, there was a corresponding difference in the cells’ storage modulus. Using flow cytometry, we measured the overall cellular stiffness using a cell cytometry device, see Appendix A. The results of the flow cytometry experiments include both the contribution from the cytoplasm and the contribution from the cellular cortex and membrane, and these experiments demonstrated no significant difference between the two populations. This observation could be due to a dominating extra contribution from the cortex or nucleus or because the flow cytometry method has less sensitivity than the optical trap-based methodology. Together, these results indicate that the cells grown in these media are mechanically closely related, however, with the periphery of the Serum/LIF cultivated cells being more elastic.

The implications of these mechanical and structural differences remain to be investigated; however, we note that some biological processes may benefit from changes in the viscoelastic properties of the cell. During the early stages of development (at the eight-cell stage), the embryo undergoes a critical compaction process where cells significantly increase their cell–cell contact area and subsequently undergo the first lineage specification to form polarized cells (outer cells, trophoblasts) and unpolarized cells (inner cells). 2i/LIF cultured cells have features in common with this stage morphologically, as they form close cell–cell contacts in 3D. During the compaction process, the cell cortex tension may play an important role in regulating cell–cell contact area. Moreover, evidence has been reported that the compaction process occurs by filopodia penetrating neighboring cells to facilitate the contraction of the embryo [52]. Penetration of filopodia into neighboring cells would be favored by a cell periphery with a lower storage modulus in agreement with what we observe for 2i/LIF cultured cells.

The physical and structural differences found here add additional evidence for the importance of cellular mechanics during development. Early development involves extensive cell segregation, which has been shown to depend on both cell adhesion and cortex tension, see, e.g., Maitre et al. [53]. Here, we measured subtle differences for the viscoelasticity of the cells cultured in Serum/LIF and 2i/LIF; however, our results show that small changes in viscoelasticity may nevertheless be associated with major structural and morphological changes.

Possible mechanisms behind this subtle mechanical difference needs further study, but notably it was recently shown that periphery in ESCs was mechanically regulated through expression of Arp 2/3, formin, and capping proteins CP [8]. Varying the relative amount of capping versus polymerization factors could allow ESCs in the process of the earliest steps in differentiation to fine-tune their cortex mechanics, which could also be a regulating factor induced by distinct media conditions as cells are shifted from 2i/LIF to Serum/LIF culture conditions.

## 5. Conclusions

Here, we have shown that subtle differences in the earliest stage of differentiation, associated with different ESC culture conditions, lead to dramatic differences in the actin nano-architecture of the cell’s cortex and also with changes in the physical properties both of the individual cells and of the colonies they form. Future studies will focus on implications of these differences on cell fate and on the mechanisms governing the interplay between the dynamic actin structure, the membrane, and cell fate regulation.

## Figures and Tables

**Figure 1 cells-10-02859-f001:**
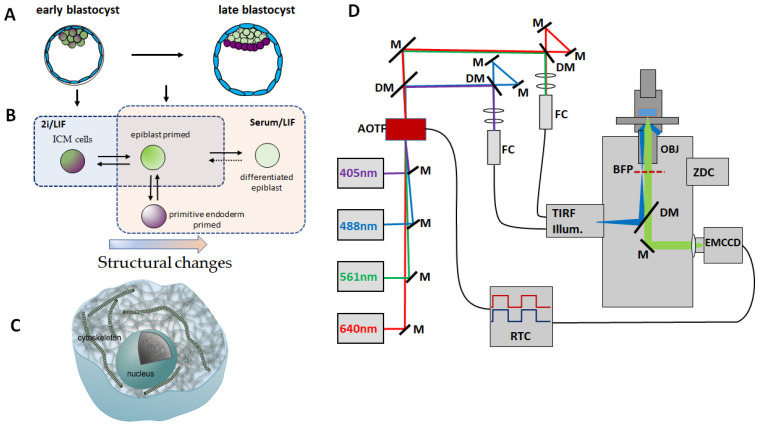
Embryonic stem cell development and the super-resolution imaging platform used to image the actin organization during early development. (**A**) Schematic illustration of the early and late stage blastocyst. (**B**) Overview of how the differentiation stage of embryonic stem cells can be modulated by culturing in different media. Cells cultured in 2i/LIF recapitulate the early preimplantation epiblast, and ICM. ESCs cultured in Serum/LIF represent a slightly later stage of development, where heterogeneous culture captures a metastable pluripotent epiblast state that exhibits reversible priming toward extra-embryonic endoderm and more differentiated epiblast [19]. (**C**) Schematics showing the cytoskeletal structure of a cell. The ultrastructure of actin is not constant during development. (**D**) Description of the TIRF/STORM setup used to image actin organization in embryonic stem cells. Optical elements and light path in the TIRF/STORM setup: *M*, mirror; *DM*, dichroic mirror; *AOTF*, acousto-optical tunable filter; *FC*, fiber coupler; *TIRF* Illumination, motorized TIRF illumination module; *BFP*, back focal plane; *OBJ***,** objective; *ZDC*, real-time z-drift compensation module; *EMCCD*, electron-multiplying CCD camera; *RTC*, real-time controller. See Appendix A for more details. Panels A and B are adapted with permission from [12].

**Figure 2 cells-10-02859-f002:**
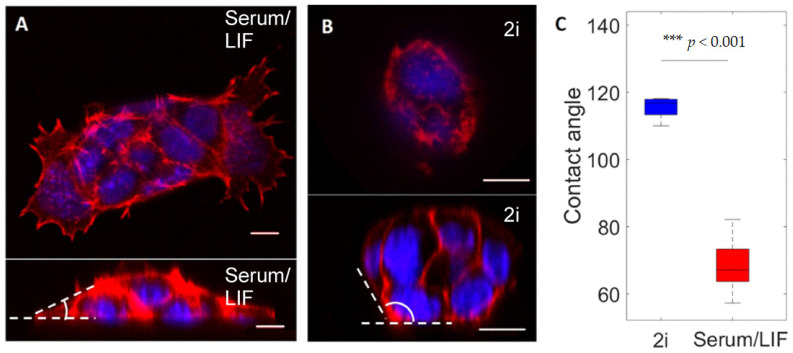
Confocal images and analysis of the shape of ESC colonies grown in 2i versus Serum/LIF. (**A**,**B**) Upper panels: images in the lateral plane (x,y) taken near the surface. Lower panels: images from a side view (x,z) of typical cell colonies of cells grown in Serum/LIF medium (**A**) or 2i medium (**B**), respectively. The contact angle of the colony to the substrate is marked in white. The scale bar is 10 μm in all images. (**C**) Contact angle between an ESC colony and the surface when cultured in 2i (blue) or in Serum/LIF (red), respectively, extracted from side-view images as shown in (**A**,**B**). *n* = 5 colonies for each condition, colonies grown in 2i exhibit significantly larger contact angles than those cultured in Serum/LIF. Error bars denote one standard deviation, and the horizontal line represents the mean values. P value was found using a standard *t*-test. *** indicates *p* < 0.001.

**Figure 3 cells-10-02859-f003:**
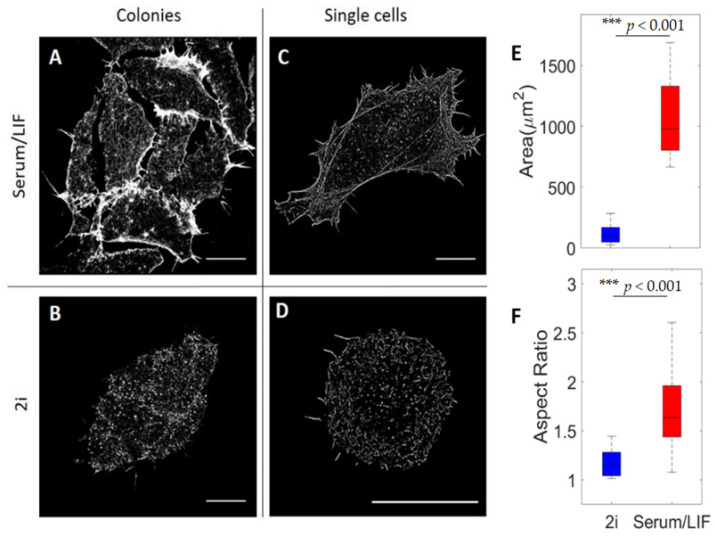
Super-resolution-based analysis of actin cytoskeleton organization in stem cells near the substrate. (**A**–**D**) STORM images of the actin network under different conditions: (**A**) a colony of cells grown in Serum/LIF media, (**B**) a colony of cells grown in 2i media, (**C**) a single cell grown in Serum/LIF media, and (**D**) a single cell grown in 2i media. All scale bars are 10 µm. (**E**) Boxplot of the surface spreading area of ESCs grown in either 2i (blue) or Serum/LIF (red) media, respectively. There is a significantly larger spreading of cells grown in Serum/LIF media than in 2i (*n* = 16, for each condition). (**F**) Boxplot of measured aspect ratios of ESCs grown in 2i (blue) or Serum/LIF (red) media, respectively (*n* = 16, for each condition). Cells cultured in Serum/LIF are significantly more elongated than those in 2i. Box edges indicate 25th and 75th percentiles, and whiskers extend to the most extreme data points not considered as outliers. *** indicates *p* < 0.001.

**Figure 4 cells-10-02859-f004:**
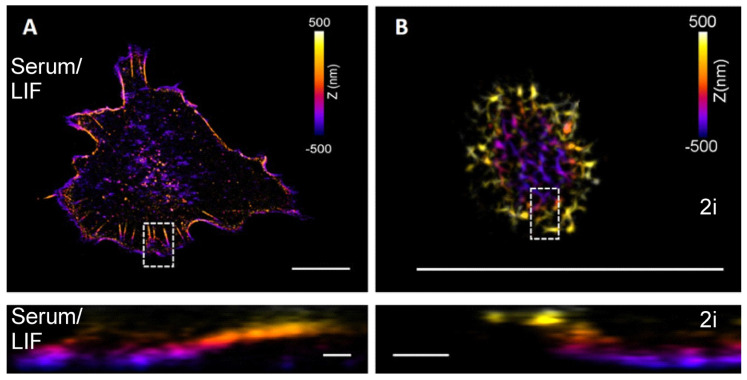
3D STORM imaging of single cells for resolving actin and stress fiber organization in ESCs grown in Serum/LIF or 2i media, respectively. (**A**) 3D visualization of actin filaments in stem cells grown in Serum/LIF. (**B**) 3D visualization of actin filaments of cells cultured in 2i. The z-positions are color-coded (violet indicating the substrate). Lower panels show side views of the boxed regions in the images above. The images indicate a dorsal type of stress fibers extending out from the surface at the edge of the cells to be present (only) in cells grown in Serum/LIF. In contrast, actin fibers in cells grown in 2i appear to extend away from the substrate close to the cell’s periphery. Scale bars in upper panels 10 μm, in lower panels 500 nm.

**Figure 5 cells-10-02859-f005:**
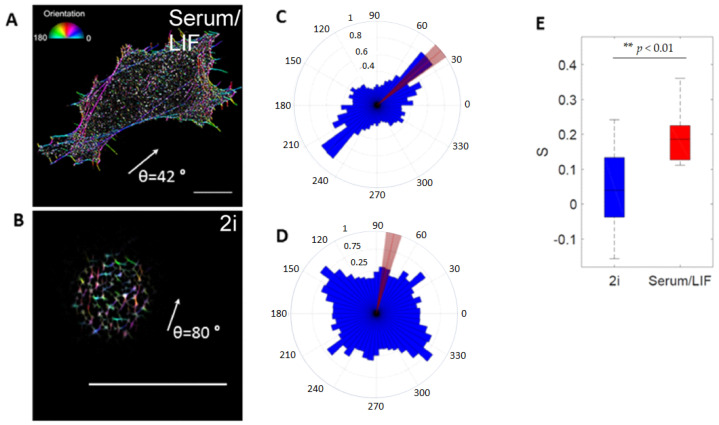
Characterization of actin filament orientation in ESCs near the surface. (**A**,**B**) Orientation of actin filaments in ESCs, color code indicates orientation with respect to the cell’s major axis (white arrow), (**A**) shows an ESC grown in Serum/LIF and (**B**) an ESC grown in 2i media. Scale bars: 10 µm. (**C**,**D**) Angular plots showing the direction (blue) of the actin filaments with (**C**) corresponding to image (**A**), and (**D**) corresponding to image (**B**), respectively. The red lines indicate the orientation of the cells’ major axes found by fitting an ellipse to the adherent area. (**E**) Boxplot of the actin filament orientation order parameter S=cos⟨(2θ)⟩, where θ is the angle between fiber orientation and the orientation of the cell’s major axis, Box edges indicate 25th and 75th percentiles, and whiskers extend to the most extreme data points not considered outliers. *n* = 16 cells for each of the 2i (blue) and Serum/LIF (red) categories. ** indicates *p* < 0.01 using *t* test.

**Figure 6 cells-10-02859-f006:**
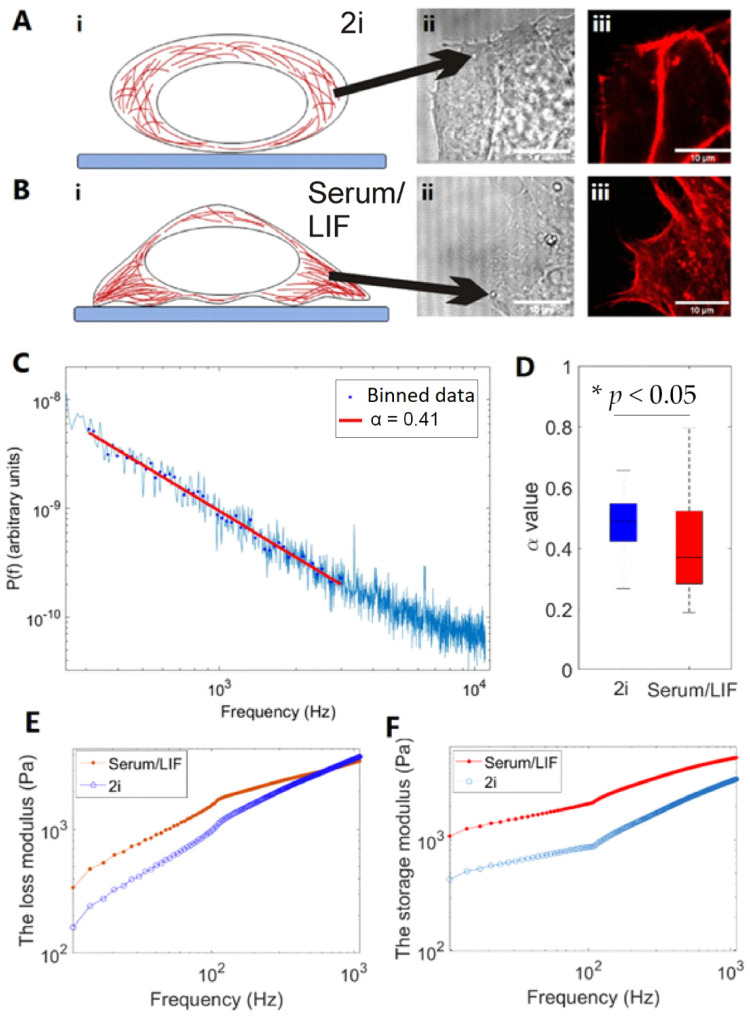
Characterization of viscoelastic properties of ESCs grown in 2i or Serum/LIF at the sub-cellular level close to the cortex of the cell. (**A**,**B**) (i) Schematic of observed cell geometry, (ii) bright field image, trapped granule marked by black arrow, (iii) confocal images of same area as shown in (ii) with F-actin labeled by a SIR-actin. In (**A**), cells were grown in 2i; in (**B**), cells were grown in Serum/LIF. (**C**) Representative power spectrum of the positions visited by a lipid granule close to the periphery of an ESC cultured in Serum/LIF. The red line shows the fit of Equation (1) to data in the frequency interval 300–3000 Hz, returning α = 0.41 for this experiment. (**D**) α-values from lipid granule trapping measurements in 2i or Serum/LIF, respectively. Box plot of 25th to 75th percentile, * indicates *p* < 0.05 using *t*-test. *n* = 26 cells for each condition. (**E**) Loss modulus, *G′*, for cells cultivated in Serum/LIF or 2i/LIF. (**F**) Storage modulus, *G″,* of cells cultivated in Serum/LIF or 2i/LIF. The magnitude of the storage modulus is significantly higher for Serum/LIF cultivated cells than for cells cultivated in 2i/LIF for all investigated frequencies, indicating that primed cells are more elastic, thus confirming the results from (**D**).

## Data Availability

Data are available upon request.

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
