# Peer review of "Changes in Cell Morphology and Actin Organization in Embryonic Stem Cells Cultured under Different Conditions"

_cells, 2021, doi:10.3390/cells10112859_

Round 1

Reviewer 1 Report

In the manuscript: “Changes in cell morphology and actin organization in embryonic stem cells cultured under different conditions”,  by Hvid et al, the authors compare and contrast ESC colonies grown in 2i versus Serum/LIF conditions, which mimic early and later stages of development. They find that 2i treated cells are more round (less spread), present sparse actin meshes and higher viscous response. By contrast, cells in Serum/LIF conditions are more spread in 2D, present actomyosin bundles and elastic mechanical response. The study is interesting and well presented. The methods used are state-of-the-art and appropriate for the type of observations and analysis they do. However, the lack of mechanistic insights makes it difficult to understand why cells at different stages of development present the reported differences. I invite the authors to provide information about possible mechanisms underlying the differences in morphology, microstructure and mechanics of the cells in these different conditions. While the connection between cell shape and cytoskeleton microstructure is intuitive, it’s not clear to me how this microstructure relates to either the viscous or elastic dominated mechanical responses. I invite the authors to propose molecular mechanisms for these dynamics. If needed, please provide more experimental data and/or evidences from previous studies to support the proposed mechanism(s).

Reviewer 2 Report

In the study entitled "Changes in cell morphology and actin organization in embryonic stem cells cultured under different conditions" by Hvid et al authors analyzed rheology, morphology and organization of the cytoskeleton in embryonic stem cells resembling different stages of development. The study is very interesting, using state of the art methods  and tackles an important point in development. Yet, some important issues remain un-addressed that are described in more detail below.

1) Authors mainly associated mechanical properties measured with the actin cytoskeleton but did not adress the influence of intermediate filaments, such as e.g. vimentin or nestin. Intermediate filaments are well known to strongly impact cell mechanics. Especially nestin might be of interest, as it was also shown to be expressed near/at the cortex of embryonic stem cells (ESC). Thus, authors should add analysis of at least nestin and potentially vimentin structure/organization in their stem cell populations (compare to figure 4/5).

2) Authors analyzed mainly single cell properties. Yet, during development, even in early stages cells form packed masses that interact with one another, influencing organization of the cytoskeleton, cell shape etc. How do the measured single cell properties relate to that? To my opinion it would strongly improve the arguments of the authors if they did additional rheological measurements of cell clusters.

3) The method used for analyzing if cells are more elastic or viscous is rather indirect and does not give absolute values of the loss or storage modulus. Looking at the presented images of the actin organization I would strongly suspect that the ESC not only have different ratios of storage to loss modulus but also very different absolute values. I recommend that authors additionally employ a more direct method for measuring storage and loss modulus. See e.g. "A one-step procedure to probe the viscoelastic properties of cells by Atomic Force Microscopy", Scientific Reports, Chim et al, 2018

4) Authors discussed the effect of cortical tension quite lengthy, with several speculations of relations between the made observations and previous studies measuring cortical tension of ESC, partly done using different conditions. To my opinion it would strongly improve the arguments of the authors if they did additional measurements of cortex tension for both culture conditions.

5) To my optionion authors should elaborate more on how the observations made may relate to the actual development.

Minor issues:

1) Please define the abbreviations 2i and LIF on first usage.

2) For all images and plots were it was not done: Please indicate the culture condition in the figure directly.

3) 2i/LIF ESC appear to adhere less to the substrate compared to Serum/LIF cells (less wetting, roundish shape, no prominent actin structures). Do these observations persist for longer cultivation times of the 2i/LIF cells, like 2-3 days after seeding on fibronectin?

4) Figure 5 is located before figure 4.

Round 2

Reviewer 1 Report

The authors have addressed all my concerns

Author Response

We thank the reviewer for the comments and for accepting the new version.

Reviewer 2 Report

The authors revised the manuscript, significantly improving it.

All issues raised by this reviewer have (mostly) been adressed and only minor points remain:

1) Please explain the flow cytometry experiments and the data-analysis in the methods section.

2) Authors noted that the rheological measurements were not taken exactly in/at the actin cortex but rather close by. I thank the authors for correcting me at this point. As mentioned in the previous comments several other important structures contributing to mechanical propreties might be located there, such as e.g. nestin. Please, briefly mention such potential explanations in the discussions.

Author Response

Reviewer 2

We thank the reviewer for the suggestion to include details concerning the methods in the manuscript and not only in the Supplementary as we did in the last revision.

We have included the following section in the materials and method section:

New text:

2.8. Real-time deformability cytometry

To assess the overall stiffness of cells we use real-time deformability cytometry where cells are flown through a channel constriction in a microfluidic device and experience deformation by pressure gradients and shear stresses, see Supplementary Fig. 3A [44, 45]. Cells were passing through a 20 μm channel cross-section. To ensure the success of the experiments, the diameter of the studied cells has to fall between 20 - 95% of the channel size.

When pumped into the chip, cells initially passed between two rows of pillars to prevent cell debris from blocking the flow through the channel. There are two inlets entering the chip containing buffer (sheath) and the sample containing the cells, see Supplementary Fig. 3A. The buffer is named Cell Carrier and is a phosphate buffered saline solution with < 1% methyl cellulose.

In the constriction, the deformed cell is illuminated with a pulsed and high-power light emitting diode and a complementary metal oxide semiconductor (CMOS) camera is used for detection. The flow cytometry arrangement combined with real-time analysis of the cell shape allows for measurement at  rates of up to 1000 cells per second.

-----

Concerning the second note about the involvement of nestin, we have included the following sentence about the possible role of nestin’s in the observed changes of the viscoelasticity.

New text:

We note that also other structures like intermediate filaments (e.g. nestin) could be partially responsible for the observed changes, but this remains to be studied in future work.